# Non-Perfusion Area Index for Prognostic Prediction in Diabetic Retinopathy

**DOI:** 10.3390/life12040542

**Published:** 2022-04-06

**Authors:** Yoshiko Ofuji, Yusaku Katada, Yohei Tomita, Norihiro Nagai, Hideki Sonobe, Kazuhiro Watanabe, Hajime Shinoda, Yoko Ozawa, Kazuno Negishi, Kazuo Tsubota, Toshihide Kurihara

**Affiliations:** 1Department of Ophthalmology, Keio University School of Medicine, 35 Shinanomachi, Shinjuku-ku, Tokyo 160-8582, Japan; ofuji.yoshiko413@keio.jp (Y.O.); yusakukatada@z2.keio.jp (Y.K.); yohei.tomita@childrens.harvard.edu (Y.T.); nagai@a5.keio.jp (N.N.); betty_vol2@ybb.ne.jp (H.S.); gaku047nikoniko3mickey@yahoo.co.jp (K.W.); shinoha@keio.jp (H.S.); ozawa@a5.keio.jp (Y.O.); kazunonegishi@keio.jp (K.N.); tsubota@tsubota-lab.com (K.T.); 2Laboratory of Photobiology, Keio University School of Medicine, 35 Shinanomachi, Shinjuku-ku, Tokyo 160-8582, Japan; 3Department of Ophthalmology Boston Children’s Hospital, Havard Medical School 3 Blackfan Circle, Center for Life Science Building, Room 18030-9, Boston, MA 02115, USA; 4Tsubota Laboratory, Inc., 34 Shinanomachi, Shinjuku-ku, Tokyo 160-0016, Japan

**Keywords:** diabetic retinopathy, fundus fluorescent, non-perfusion area

## Abstract

Fundus fluorescent angiography is a standard examination in Japan that can directly visualize the circulatory failure in diabetic retinopathy but is not used in Western countries. In this study, we examine the relationship between the non-perfusion area in fundus fluorescent angiography and the progression of diabetic retinopathy. We evaluated 22 eyes between 22 patients who had their first fundus fluorescent angiography during a clinical episode at Keio University Hospital from January 2012 to May 2015, were diagnosed as having preproliferative diabetic retinopathy, and could be followed for at least three years. The non-perfusion area index (%) in nine segmented fundi in the initial fundus fluorescent angiography was calculated, and the progression to proliferative diabetic retinopathy over three years was evaluated. Three out of the 22 eyes (13.6%) developed proliferative diabetic retinopathy over three years. The non-perfusion area index for the initial fundus fluorescent angiography was significantly associated with progression to proliferative diabetic retinopathy. The non-perfusion area index in the posterior pole was most strongly correlated with the progression to proliferative diabetic retinopathy. Thus, the non-perfusion area index in the posterior pole among those with preproliferative diabetic retinopathy may predict the progression to proliferative diabetic retinopathy in the subsequent three years.

## 1. Introduction

Fundus fluorescent angiography (FA) is a useful examination to detect insufficient retinal perfusion, the essential pathological component of diabetic retinopathy (DR).

FA is not very prevalent in Western countries in the management of diabetic retinopathy. The Early Treatment Diabetic Retinopathy Study (ETDRS) conducted in the 1980s suggested that although FA was useful in predicting the prognosis of DR, it was not necessary to predict the progression to proliferative diabetic retinopathy (PDR) under ophthalmology at that time. Hence, ETDRS concluded that color stereoscopic fundus photographs alone were sufficient for DR management [1]. Based on this impactful study, FA has not been the standard practice for DR in Western countries for a long period of time. In fact, even the recently updated guidelines for DR in the United States maintain that FA should be done only when appropriate and offers no active recommendations [2], and that routine FA is not indicated as a part of the regular examination of patients with diabetes [3].

On the other hand, FA is widely adopted in Japan. The non-perfusion area (NPA) is confirmed using FA in patients with DR, and selective photocoagulation or panretinal photocoagulation is performed according to the extent of the NPA [4]. In clinical settings, advanced retinopathy is often detected even without apparent abnormal findings on fundus photographs [1]. It can, therefore, be inadequate to manage DR based on fundus examination alone.

In this context, we aimed to re-evaluate the efficacy of FA for DR management by examining the association between the NPA detected by FA and retinopathy progression. Now that 30 years have passed since ETDRS and the number of treatment options has increased [2], it is possible that the availability of FA has also increased [5].

## 2. Materials and Methods

Among the 129 patients with DR who had their first FA during a clinical episode at Keio University Hospital from January 2012 to May 2015 and were diagnosed as having preproliferative diabetic retinopathy (PPDR) according to the modified Davis classification [6], we included 22 eyes from 22 patients that could be followed with FA for at least three years (Keio University School of Medicine Ethics Committee approval number 20170049). The right eye was included in all cases, and cases of unclear fundus due to corneal and intermediate translucent opacity were excluded. In addition, cases where patients could not obtain a fluorescent fundus image for some reason, such as side effects during examination, and cases where patients could not obtain a fundus image for some reason, such as poor fixation or difficulty in maintaining posture, were also excluded. There was no other exclusion criterion for additional systemic or ocular problems such as retinal vein occlusion that can cause NPA.

For each of the 22 eyes, we calculated the non-perfusion area index (NPAI; %) for each of the nine quadrants illustrated in Figure 1 to represent the ratio of NPA (pixels) to the total imaging area (pixels) as a percentage.

The fundus was divided into nine quadrants. The angle of view for each quadrant of FA we used was 50 degrees. Two ophthalmologists familiar with the interpretation of FA for diabetic retinopathy manually surrounded the non-perfusion area (NPA) using the lasso tool in Photoshop and calculated it as the number of pixels. They calculated the percentage of the NPA (pixels) in the total imaging area (pixels) for each of the nine quadrants, respectively, and we used the average of the two as data.

Using medical records, we extracted patients’ characteristics such as age, sex, height, weight, history of diabetes, insulin use, HbA1c, blood pressure, body temperature, best corrected visual acuity (BCVA; logMAR), refraction, intraocular pressure, axial length, presence of macular edema, and NPAI at the time of the initial FA. For HbA1c, BCVA, refraction, intraocular pressure, and the presence of macular edema, we also obtained the data at the time of the final FA (mean 41.10 ± 5.68 months after the initial FA). We extracted information on the intervention (photocoagulation, the number of anti-vascular endothelial growth factor (VEGF) drugs administration, and vitrectomy for DR) and progression to PDR at the time of the final FA.

Three years after the initial FA, the risk factors for progression to PDR were retrospectively examined.

## 3. Results

Patient background (Table 1) and NPAI at the initial FA (Table 2) are shown. When comparing the NPAIs in each quadrant, there was no apparent significant difference between the quadrants, but there was a tendency for NPAIs to be larger in the inferior nasal quadrants.

BCVA tended to decrease from the initial to the final FA, but the difference was not statistically significant, and macular edema was significantly reduced (Table 3). Three of the 22 eyes (13.6%) were transferred to PDR. A total of 13 cases underwent photocoagulation, including all three cases that had progressed to PDR (Table 4 and Table 5). Vitrectomy was performed on one case for diabetic macular edema with epiretinal membrane that did not improve even after three administrations of anti-VEGF drug.

The non-progressed group included cases in which HbA1c improved significantly after being hospitalized for diabetic education. There were no significant differences in the history of diabetes, insulin use, HbA1c, or BCVA between the PDR-progressed and non-progressed groups. Only the mean value of the nine-quadrant NPAIs at the initial FA was significantly different and was significantly larger in the progressed group (Table 5). When comparing the NPAIs in each quadrant between the two groups, the NPAIs in six of the nine quadrants, except for the superior temporal, superior, and superior nasal quadrants, were significantly larger in the progressed group (Table 6 and Table 7).

Correlation coefficients were calculated using the presence or absence of progression to PDR during the observation period as a binary variable, and NPAI in the posterior pole during the initial FA showed the strongest correlation (Table 8).

## 4. Discussion

In this study, we evaluated the association between the non-perfusion area detected by FA and retinopathy progression to examine the usefulness of FA in DR practice. The mean of the nine-quadrant NPAI at the initial FA was significantly associated with PDR progression in the modified Davis classified PPDR, indicating the capability of FA to predict the progression of DR.

The NPAIs in six of the nine quadrants, except for the superior temporal, superior, and superior nasal quadrants, were significantly larger in the progressed group, and NPAI in the posterior pole showed the strongest correlation with PDR progression. These results suggest that NPAI in the posterior pole at the initial FA may be helpful to predict the progression from PPDR to PDR.

Compared to the previous evidence, our results indicate that the treatment using FA may have been able to further suppress the progression of PPDR to PDR. The studies in Western countries, where FA was not used for general management, reported that 12–27/52/75% [1] of moderate/severe/very severe non-proliferative diabetic retinopathy (NPDR) patients developed PDR within one year, and 27.2/45.5% [7] of moderate/severe NPDR developed PDR in five years. Those figures are higher than the PDR progression rate in this study, which was 13.6%. The Japan Diabetes Complications Study (JDCS), a large randomized controlled trial [8], found that the PDR progression rate tended to be lower than that in other countries, suggesting that FA-oriented DR care was effective in preventing progression to PDR. Another Japanese study likewise reported that the progression to PDR in three years was 22% [9], which is consistent with the results of this study.

Some previous reports showed that NPA was dominant in the mid-periphery [10,11,12], while another report showed that NPA was dominant in the nasal side, which is consistent with our results. Furthermore, Niki et al. reported that NPA in the posterior pole was more likely to predict future progression to PDR [10], which is consistent with our results, whereas NPAI in the posterior pole was strongly correlated with the PDR transition. Although further comprehensive evaluations are needed because the mapping of the NPA area differs from one study to another, our study adds additional findings to the literature. It should be noted that the definition of the NPA area slightly differs between reports.

For PPDR cases, FA should be actively performed. Moreover, especially for cases with a large amount of NPA in the posterior pole, precautious photocoagulation or anti-VEGF drug should be conducted in a proactive manner. OCTA may be useful in understanding the NPA in the posterior pole. In addition, from the viewpoint that the prognosis can be predicted by the amount of NPA in the posterior pole, it may lead to alerting patients and encouraging medical treatment.

In this study, anti-VEGF was administrated to eight of 11 cases of macular edema, and two of three cases that were not injected progressed to PDR. The prompt administration of anti-VEGF drug is suggested for cases with macular edema, and it is possible that anti-VEGF suppresses the progression to PDR. In addition, we performed photocoagulation on 13 cases, of which 10 cases did not progress to PDR. By performing photocoagulation on NPA detected by FA before the appearance of neovascularization, it may be able to suppress the progression to PDR.

There are several limitations to this study. First, the number of cases was small. Further investigations are, therefore, needed to confirm our results. Furthermore, this study was retrospective, and the patient background and conditions of the treatment intervention were not adjusted completely. Even with these limitations, our report is valuable because we presented the potential capability of FA to predict the progression of PPDR to PDR.

## 5. Conclusions

In conclusion, three of the 22 patients (13.6%) developed PDR during a mean follow-up of 41.1 months. The mean of the nine-quadrant NPAI at the initial FA was significantly associated with PDR progression in the modified Davis classified PPDR, suggesting the capability of FA to predict the progression of PPDR to PDR.

## Figures and Tables

**Figure 1 life-12-00542-f001:**
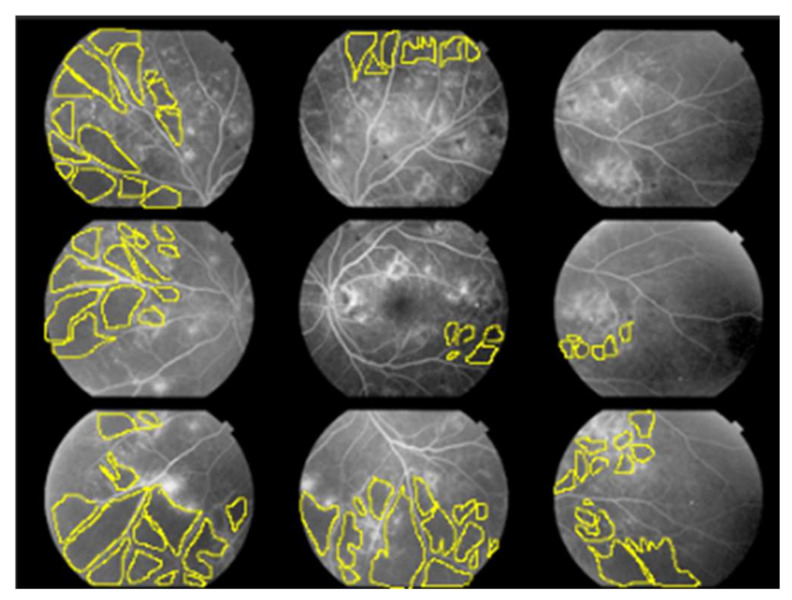
Non-perfusion area index (NPAI); %.

**Table 1 life-12-00542-t001:** Patient background.

**Age (years)**	59.5 ± 11.6(38–76)
**Gender**	17 men, 5 women
**BMI (kg/m^2^)**	27.1 ± 14.6 (19.4–39.1)
**History of diabetes (years)**	11.7 ± 9.0 (0–30)
**Whether or not insulin was used (yes:1, no:0)**	0.5 ± 0.503
**HbA1c (%)**	7.97 ± 2.25 (6.0–12.0)
**Systolic/Diastolic blood pressure (mmHg)**	130.2 ± 21.7 (101–190)/75.4 ± 12.1 (57–104)
**Body temperature (°C)**	36.5 ± 0.5 (35.5–37.2)
**Corrected visual acuity (logMAR)**	0.108 ± 0.258 (−0.079–1.000)
**Equivalent spherical power (D)**	−2.39 ± 4.43 (−13.63–2.13)
**Intraocular pressure (mmHg)**	14.3 ± 2.7 (9–19)
**Axial length (mm)**	24.3 ± 1.2 (23.09–26.51)
**Presence (1) or absence (0) of macular edema (%)**	57.9 ± 51.2
**NPAI (%)**	15.9 ± 25.5 (0.01–77.7)

Mean ± standard deviation, BMI = body mass index, NPAI = non-perfusion area index.

**Table 2 life-12-00542-t002:** NPAI (%) in each quadrant at initial FA.

Upper temporal 10.3 ± 26.4 (0–93.6)	Upper 12.2 ± 21.6 (0–65.3)	Upper nasal 17.6 ± 30.3 (0–85.3)
Temporal 13.1 ± 27.1 (0–100)	Post pole 8.0 ± 14.5 (0–37.0)	Nasal 20.9 ± 31.8 (0–97.9)
Lower temporal 17.4 ± 32.5 (0–100)	Lower 21.1 ± 33.4 (0–90.3)	Lower nasal 22.4 ± 36.3 (0–100)

Mean ± standard deviation *p* > 0.05 ANOVA, NPAI = non-perfusion area index, FA = fundus fluorescent angiography.

**Table 3 life-12-00542-t003:** Comparison of survey items at initial FA and final FA (Mean 41.10 ± 5.68 months later).

	At Initial FA Mean ± Standard Deviation	At Final FA Mean ± Standard Deviation	*p* Value
**HbA1c (%)**	7.97 ± 2.25 (6.0–12.0)	7.59 ± 3.23 (5.4–9.7)	>0.05
**Corrected visual acuity (logMAR)**	0.108 ± 0.258 (−0.079–1.000)	0.165 ± 0.376 (−0.079–1.222)	>0.05
**Equivalent spherical degree (D)**	−2.39 ± 4.43 (−13.63–2.13)	−2.00 ± 4.03 (−13.63–2.50)	>0.05
**Intraocular pressure (mmHg)**	14.3 ± 2.7 (9–19)	14.0 ± 2.3 (10–17)	>0.05
**Existence of macular edema**	0.579 ± 0.512	0.500 ± 0.512	<0.01 *

* Wilcoxon signed-rank test and χ^2^ test, FA = fundus fluorescent angiography.

**Table 4 life-12-00542-t004:** Treatment interventions during the observation period.

	Mean ± Standard Deviation
**Existence of photocoagulation (%)**	13 cases (59.1%)
**Existence of anti-VEGF drug administration (%)**	8 cases (36.4%)
**Number of administrations of anti-VEGF drug (times)**	3.32 ± 5.5 (0–16)
**Existence of vitreous surgery for DR (%)**	1 case (4.55%)

VEGF = vascular endothelial growth factor, DR = diabetic retinopathy.

**Table 5 life-12-00542-t005:** Comparison of PDR non-transition group and transition group.

	Non-Progressed Group (*n* = 19) Mean ± Standard Deviation	Progressed Group (*n* = 3) Mean ± Standard Deviation	*p* Value
**Age (year)**	60.1 ± 11.8 (38–76)	56.0 ± 11.8 (43–66)	0.619
**Weight (kg)**	76.5 ± 22.2 (53.4–131.0)	50.0 ± 0.0 (50)	0.182
**Diabetes history** **at initial FA (year)**	11.56 ± 9.39 (0–30)	12.33 ± 4.04 (10–17)	0.819
**Existence of insulin use** **at initial FA**	9/16 cases (56.3%)	0/2 cases (0%)	0.298
**HbA1c at initial FA (%)**	7.80 ± 1.30 (6.0–11.0)	8.97 ± 2.63 (7.4–12.0)	0.524
**Corrected visual acuity** **at initial FA (logMAR)**	0.51 ± 0.17 (−0.08–0.40)	0.47 ± 0.47 (0.10–1.00)	0.267
**Existence of macular edema** **at initial FA**	8/16 cases (50%)	3/3 cases (100%)	0.107
**NPAI of nine quadrant average** **at initial FA (%)**	10.3 ± 21.9 (0.01–77.7)	50.5 ± 19.2 (30.1–64.9)	0.021 *
**Existence of photocoagulation**	10/19 cases (52.6%)	3/3 cases (100%)	0.121
**Number of administrations of** **anti-VEGF drug (times)**	3.21 ± 5.45 (0–16)	4.00 ± 6.93 (0–12)	0.865
**Existence of vitreous surgery** **for DR (%)**	0.05 ± 0.23	0.00 ± 0.00	0.331
**HbA1c at final FA (%)**	7.71 ± 1.4 (5.4–9.7)	7.00 ± 1.3 (5.5–8.0)	0.574
**Corrected visual acuity** **at final FA (logMAR)**	0.13 ± 0.31 (−0.08–1.05)	0.41 ± 0.71 (−0.08–1.22)	0.556
**Existence of macular edema** **at final FA**	10/19 cases (52.6%)	1/3 cases (33.3%)	0.534

Student’s *t*-test * *p* < 0.05, PDR = proliferative diabetic retinopathy, FA = fundus fluorescent angiography, NPAI = non-perfusion area index, VEGF = vascular endothelial growth factor, DR = diabetic retinopathy.

**Table 6 life-12-00542-t006:** NPAI at initial FA in non-progressed group (%).

Upper temporal 10.0 ± 28.3 (0–93.6)	Upper 8.2 ± 17.6 (0–56.7)	Upper nasal 13.8 ± 28.3 (0–85.3)
Temporal 10.0 ± 27.7 (0–100)	Post pole 2.9 ± 5.1 (0–19.3)	Nasal 12.4 ± 23.4 (0–94.4)
Lower temporal 10.8 ± 27.9 (0–100)	Lower 11.7 ± 24.4 (0–72.5)	Lower nasal 13.6 ± 29.2 (0–100)

Mean ± standard deviation Student’s *t*-test * *p* < 0.05, NPAI = non-perfusion area index, FA = fundus fluorescent angiography.

**Table 7 life-12-00542-t007:** NPAI at initial FA in progressed group (%).

Upper temporal 12.4 ± 10.0 (3.6–24.0)	Upper 37.7 ± 31.9 * (2.8–65.3)	Upper nasal 41.9 ± 37.5 (4.4–79.4)
Temporal 33.6 ± 10.7 (23.7–44.9)	Post pole 36.0 ± 19.5 * (29.0–55.1)	Nasal 74.8 ± 25.3 * (47.7–97.9)
Lower temporal 59.3 ± 32.8 * (23.1–86.9)	Lower 80.5 ± 15.4 * (62.7–90.3)	Lower nasal 78.4 ± 26.8 * (48.4–100)

Mean ± standard deviation Student’s *t*-test * *p* < 0.05, NPAI = non-perfusion area index, FA = fundus fluorescent angiography.

**Table 8 life-12-00542-t008:** Correlation between PDR progression and NPAI at initial FA.

Upper temporal 0.033	Upper 0.478 *	Upper nasal 0.326
Temporal 0.305	Post pole 0.904 **	Nasal 0.689 **
Lower temporal 0.523 *	Lower 0.724 *	Lower nasal 0.627 **

Adapted as a binary variable with/without PDR progression, Pearson correlation coefficient * *p* < 0.05, ** *p* < 0.01, PDR = proliferative diabetic retinopathy, NPAI = non-perfusion area index, FA = fundus fluorescent angiography.

## Data Availability

The data presented in this study are available upon request from the corresponding author.

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
