# Peer review of "Non-Perfusion Area Index for Prognostic Prediction in Diabetic Retinopathy"

_life, 2022, doi:10.3390/life12040542_

Round 1

Reviewer 1 Report

Fundus fluorescent angiography is a useful examination for diagnosis of diabetic retinopathy. In this study, the author performed a retrospective analysis to examine the correlation between NPAI and DR progression. However, the author should polish the writing and add more detail to prove the authenticity of the result. Specific comments are as below:

  1. Line 40,41: FA is a widely recognized tool especially for preproliferative diabetic retinopathy patients. The reference 1 and 2 published in 1991, 2017 respectively are outdated. It is recommended to refer the latest literature.
  2. Method: Please indicate the scope of FA, 60°, 120° or 200°
  3. Method: How was the nonperfusion area identified? Manually outline? What was the criteria used for nonperfusion area?
  4. In Table 5: Firstly, there should be 19 patients in non-progressed group, however, the data below just presented 16 patients’ information, where are the other 3 patients? Secondly, it’s intriguing that there is 1 vitreous surgery case in non-progressed group other than progressed group, why is that? Thirdly, the HbA1c of progressed is better than non-progressed group at final FA, nevertheless it’s obviously worse at first FA, why is that?
  5. Discussion section: It’s intervention such anti-VEGF and photocoagulation prevents the progression of DR rather than FA, it’s suggested to rephrase.
  6. It’s obvious that the use of anti-VEGF and photocoagulation may influence the progression of DR, please add more discussion on this. For example, precautious photocoagulation will change the progression of DR.

Reviewer 2 Report

This is an article entitled “Non-perfusion area index for prognostic prediction in diabetic retinopathy (life-1615546)” which evaluates the relationship between the non-perfusion area in fundus fluorescent angiography and the progression of diabetic retinopathy.

Abstract

  •  

Introduction

  •  

Materials&Methods

  • The eye number seems to be insufficient.
  • Were there any additional systemic or ocular problems of the patients? Please add the exclusion criterion in more detail.

Results

  • Please add the ranges of all data.

Discussion

  • Should be enlargened to discuss the possible effects of the treatment modalities used for diabetes.

Tables

  • Please add the ranges of all data.

References

  • Ok

Round 2

Reviewer 1 Report

The authors have answered my questions. Thank you.

Reviewer 2 Report

This is an article entitled “Non-perfusion area index for prognostic prediction in diabetic retinopathy (life-1615546)” which evaluates the relationship between the non-perfusion area in fundus fluorescent angiography and the progression of diabetic retinopathy.

The major limitation of the study is the limited number of eyes.